# Magnesium Fluoride Forms Unique Protein Corona for Efficient Delivery of Doxorubicin into Breast Cancer Cells

**DOI:** 10.3390/toxics7010010

**Published:** 2019-02-22

**Authors:** Hamed Al-Busaidi, Md. Emranul Karim, Syafiq Asnawi Zainal Abidin, Kyi Kyi Tha, Ezharul Hoque Chowdhury

**Affiliations:** 1Jeffrey Cheah School of Medicine and Health Sciences, Monash University Malaysia, Jalan Lagoon Selatan, Bandar Sunway, Subang Jaya 47500, Malaysia; halb0005@student.monash.edu (H.A.-B.); karim604306@gmail.com (M.E.K.); syafiqnawi@gmail.com (S.A.Z.A.); tha.kyi.kyi@monash.edu (K.K.T.); 2Health & Wellbeing Cluster, Global Asia in the 21st Century (GA21) Platform, Monash University Malaysia, Jalan Lagoon Selatan, Bandar Sunway, Subang Jaya 47500, Malaysia

**Keywords:** MgF_2_, nanoparticle, drug delivery, cellular uptake, cytotoxicity, opsonin, dysopsonin, pH-sensitive, drug release, protein corona

## Abstract

Background: The efficacy of chemotherapy is undermined by adverse side effects and chemoresistance of target tissues. Developing a drug delivery system can reduce off-target side effects and increase the efficacy of drugs by increasing their accumulation in target tissues. Inorganic salts have several advantages over other drug delivery vectors in that they are non-carcinogenic and less immunogenic than viral vectors and have a higher loading capacity and better controlled release than lipid and polymer vectors. Methods: MgF_2_ crystals were fabricated by mixing 20 mM MgCl_2_ and 10 mM NaF and incubating for 30 min at 37 °C. The crystals were characterized by absorbance, dynamic light scattering, microscopic observance, pH sensitivity test, SEM, EDX and FTIR. The binding efficacy to doxorubicin was assessed by measuring fluorescence intensity. pH-dependent doxorubicin release profile was used to assess the controlled release capability of the particle-drug complex. Cellular uptake was assessed by fluorescence microscopy. Cytotoxicity of the particles and the drug-particle complex were assessed using MTT assay to measure cell viability of MCF-7 cells. Results and Discussion: Particle size on average was estimated to be <200 nm. The crystals were cubic in shape. The particles were pH-sensitive and capable of releasing doxorubicin in increasing acidic conditions. MgF_2_ nanocrystals were safe in lower concentrations, and when bound to doxorubicin, enhanced its uptake. The protein corona formed around MgF_2_ nanoparticles lacks typical opsonins but contains some dysopsonins. Conclusion: A drug delivery vector in the form of MgF_2_ nanocrystals has been developed to transport doxorubicin into breast cancer cells. It is pH-sensitive (allowing for controlled release), size-modifiable, simple and cheap to produce.

## 1. Introduction

Breast cancer treatment constitutes a large proportion of worldwide health expenditure. In the US, it costs an average of $537 per person per month [1]. In addition, chemotherapy-related adverse effects cost $1271 per person per year and ambulatory encounters $17,617 per person per year [2]. The cost of care for breast cancer could be significantly reduced by employing a drug delivery system to reduce the administered dose of the drug and off-target side effects.

Using nanoparticles as drug delivery agents has gained increased interest to researchers due to several advantages, such as targeting cancer cells, reducing off-target effects, reducing systemic clearance of the drug, increasing drug uptake into the cell, capacity to deliver a wide range of therapeutics including siRNA and DNA, and combining drug delivery with other modes of treatment e.g., hyperthermia [3] and imaging [4].

However, there are some issues related to the use of nanoparticles. These include accumulation within liver, spleen, kidney and reticuloendothelial system, and difficulties with targeting cancer cells, since nanoparticles rely on the availability of adequate circulation (to accumulate in cancer tissue) and lymphatic drainage (to prevent a rise in intratumoural pressure which would hamper the movement of nanoparticles). To address these problems, different types of nanoparticles have been developed and new ones still are being developed.

Inorganic salts such as Group 2 metal fluorides, carbonates, sulfates and sulfites have plenty of advantages over other drug delivery systems in that they are non-carcinogenic and less immunogenic than viral vectors, and have a higher loading capacity and better controlled release than lipid and polymer vectors. They are simple and thereby cheaper to produce, size-modifiable and can be conjugated with targeting moieties. In particular, magnesium is higher up the Group 2 metals table, making it less reactive (hence more stable) than calcium, strontium and barium particles. The ubiquity of magnesium in the human body [5] suggests that it would have less toxicity risks, by comparison to iron, gold and silver nanoparticles. Water-insoluble magnesium salt particles would thus present an excellent candidate for research on drug delivery.

Doxorubicin is a commonly studied drug in breast cancer research. It is an anthracycline drug used to treat breast cancer. Doxorubicin’s mechanism of action involves intercalating DNA directly and inhibiting its uncoiling; inhibition of topoisomerase enzymes and generating free radicals. These actions damage DNA, inhibit DNA replication and ultimately lead to cell death (by apoptosis). Doxorubicin’s use is largely limited by its cardiotoxicity [6,7]. Strategies used to address the issue of doxorubicin toxicity have included the fabrication of nanoparticle delivery systems, the development of doxorubicin analogs; the use of low dose, prolonged, continuous infusion regimens and combining doxorubicin treatment with dexrazoxane, a cardioprotective drug [8].

Nanoparticle formulations employed to deliver doxorubicin include liposomes, polymers and inorganic particles. Liposomes are rapidly cleared by the reticuloendothelial system (within a few hours) and thus have a short half-life. To fix this problem, liposomes were conjugated with polymers such as polyethylene glycol (PEG). PEGylated liposomal doxorubicin (Doxil^®^) has been approved by the food and drug administration (FDA) for treating ovarian cancer. Despite increasing the half-life of liposomal doxorubicin, PEGylation increases the risk of hand-foot syndrome, also known as palmar-plantar erythrodysesthesia [8]. Superparamagnetic iron oxide (SPION) nanoparticles can add imaging (as contrast agents for MRI) and hyperthermia functions to doxorubicin delivery. Despite this promising prospect, more studies will be needed to assess their cytotoxicity [9].

Here we report on the first use of magnesium fluoride as a drug delivery agent for doxorubicin in cancer treatment, its manufacture, binding and release under pH change, cellular uptake and cytotoxicity. Magnesium fluoride nanoparticles were successfully produced and bound to doxorubicin. Fluorescent microscopic images showed successful internalisation of the drug-bound nanoparticles by MCF-7 cells. MTT analysis showed an increased toxicity when doxorubicin was bound to MgF_2_ nanoparticles. Protein corona was analysed by LC/MS and contained a distinct lack of typical opsonins such as immunoglobulins and complement proteins, crucial for clearance by macrophages.

## 2. Materials and Methods

### 2.1. Materials Used

MgCl_2_, NaF, NaHCO_3_, 8.621 mM doxorubicin-hydrochloride (Dox) D1515, Foetal bovine serum (FBS), Trypsin-ethylenediaminetetraacetic acid (EDTA), Na_2_HPO_4_, K_2_HPO_4_, Thiazolyl blue tetrazolium bromide (MTT), dimethyl sulfoxide (DMSO), TrypLE, penicillin streptomycin, HEPES and liquid Dulbecco’s modified eagle medium (DMEM) were bought from Sigma Aldrich (St. Louis, MO, USA). DMEM was bought from Gibco by Life Technology (Thermo Fischer Scientific, Waltham, MA, USA). NaCl, KCl and HCl were bought from Fischer Scientific (Loughborough, UK). Milli-Q water purification from Millipore was used.

### 2.2. Fabrication of Nanocrystals

MgCl_2_ was fixed at 20 µL of 1 M and mixed with increasing concentrations of NaF; 10, 20, 30, 40, 50 µL of 1 M. The solution was incubated at 37 °C for 30 min, after which 10% FBS-supplemented DMEM was added to make a total of 1 mL. The pH was calibrated within the range of 7.4 to 7.55.

Formation of nanoparticles was estimated using a Jasco UV-VIS spectrophotometer (Ocklahoma city, OK, USA) to measure absorbance values at 320 nm wavelength. Samples were measured in triplicates and an average value was calculated.

The effect of incubation time on the formation of nanocrystals was determined using the same procedure, but incubating for 60 min for comparison.

### 2.3. Dynamic Light Scattering (DLS)

Particles were prepared as mentioned in Section 2.2. 

Dynamic light scattering was performed to estimate particle sizes. Measurements were made using Malvern nano zeta sizer (Worcestershire, UK) and accompanying software. Samples were measured in duplicates and an average value was calculated.

### 2.4. Microscopic Observation of Particles

Particles were prepared as mentioned in a previous section. The particles were visualised in brightfield 10× magnification. The microscope used was Olympus Fluorescence Microscope IX81 and CellSens Dimension software (Tokyo, Japan).

### 2.5. Fourier Transform Infrared Spectroscopy (FTIR)

2 mL of 1 M MgCl_2_ was added to 1 mL of 1 M NaF. The samples were then incubated at 37 °C for 30 min. 97 mL of 10% FBS-supplemented DMEM was added to make a total of 100 mL, which was then centrifuged at 3000 rpm, 4 °C for 45 min. The supernatant was then discarded, and the precipitate was freeze-dried using Labconco freeze dryer (Kansas city, MO, USA).

Varian FTIR (Santa Clara, CA, USA) and Varian Resolution Pro 640 software were used to check the spectra of the MgF_2_ particles.

### 2.6. Field Emission Scanning Electron Microscope (FE-SEM) and Energy Dispersive X-Ray Spectroscopy (EDX)

MgF_2_ particles were prepared by adding 20 µL of 1 M MgCl_2_ to 10 µL of 1 M NaF. The sample was incubated at 37 °C for 30 min. 3 µL of the sample was then left to dry for 15 h on a glass slip. The dried sample was subjected to platinum sputtering for 45 s at a current of 30 mA, and a tooling factor of 2.3. The particles were visualised at 10 kV (for EDX) and 5 kV (for SEM). The device used was Hitachi/SU8010 (Tokyo, Japan).

### 2.7. pH Sensitivity of the Nanoparticles

20 µL of 1 M MgCl_2_ was added to 10 µL of 1 M NaF and the sample was incubated at 37 °C for 30 min. DMEM with 10% FBS media was added to the sample, adjusted to pH values in the range: 7.5, 7.0, 6.5, 6.0, 5.5. The samples were then measured for absorbance using Jasco UV-VIS spectrophotometer. The samples were prepared in triplicates and a mean value was calculated.

### 2.8. Binding Efficiency of Particles to Doxorubicin

To estimate the drug binding affinity of MgF_2_ nanoparticles, doxorubicin-bound nanoparticles were generated by adding 20 µL of 1 M of MgCl_2_, 10 µL of 1M NaF and doxorubicin in varying concentrations of 5, 10, 15 and 20 μM. The samples were then incubated at 37 °C for 30 min and centrifuged at 6000 rpm, 4 °C for 20 min. The supernatant was then discarded and the precipitate resuspended in 100 µL of EDTA (5 mM) in PBS. The final samples were placed in 96 well plates.

PerkinElmer microplate reader (Waltham, MA, USA) was used to measure their fluorescence intensity, and a standard curve of doxorubicin concentration against fluorescence intensity was plotted, shown in Figure 1. The fraction of doxorubicin present in the final nanoparticle suspension divided by the concentration originally used to form the nanoparticle suspension gives us the binding affinity.

All samples were prepared in triplicates and a mean result was taken.

### 2.9. pH Release Profile of Doxorubicin

20 µL of 1 M MgCl_2_, 10 µL of 1 M NaF, 50 µL of 100 µM doxorubicin were added together. The sample was incubated at 37 °C for 30 min. The samples were then added to pH-adjusted media in the ranges of: 7.5, 7.0, 6.5, 6.0, 5.5, 5.0. Next, the samples were centrifuged at 6000 rpm, and 4 °C for 20 min. The supernatant was then discarded leaving behind 30 µL of precipitate, which was resuspended in 170 µL of 5 mM EDTA in PBS. A control sample was prepared by 30 µL of 10% FBS-supplemented DMEM media at 7.5 pH dissolved in 170 µL of EDTA (in order to subtract DMEM’s own fluorescence intensity from the samples’ values). The samples were placed in a 96 well microplate, and fluorescence intensity was then measured using PerkinElmer microplate reader. Each sample was prepared as a triplicate and a mean value calculated.

### 2.10. Cell Culture and Seeding

MCF-7 cells were incubated in a 25 cm^3^ flask in liquid DMEM media supplemented with 1% HEPES buffer media, 1% streptomycin and 10% FBS. The flask was incubated in 37 °C and 5% CO_2_. MCF-7 cells in DMEM were seeded into 24 well plates (Greiner, Germany). 50,000 cells were seeded per well. Estimations of cell counts were made using a Hirschmann counting chamber (haemocytometer). The wells were incubated in 37 °C and 5% CO_2_ for 24 h.

### 2.11. Cellular Uptake of Doxorubicin-Bound Nanoparticles

Cells were seeded as mentioned before.

The cells were then treated with MgF_2_- bound doxorubicin complexes, prepared by adding 20 µL of 1 M MgCl_2_, 10 µL of 1 M NaF and 10 µM doxorubicin; incubation at 37 °C for 30 min; and adding 10% FBS-supplemented DMEM media to make a 1 mL suspension.

The control was 10 µM doxorubicin in 1 mL of 10% FBS-supplemented DMEM media. Uptake was determined by fluorescence images taken after 2 h. Media was removed from the wells, and the wells were rinsed with 100 µL of 5 mM EDTA followed by 100 µL PBS twice. Then 100 µL PBS was added to the wells before observation. The microscope used was Olympus Fluorescence Microscope IX81 and CellSens Dimension software.

### 2.12. Cytotoxicity of MgF_2_ Particles

Cells were cultured and seeded as mentioned above. Nanoparticles were fabricated by adding 20 µL of 1 M MgCl_2_ and a range of concentrations of NaF: 10, 20, 30, 40, 50 uL of 1 M NaF, and incubated at 37 °C for 30 min. The nanoparticle samples were then centrifuged at 13,000 rpm and 20 °C for 10 min. The supernatant was discarded leaving behind 10 µL of precipitate. The precipitate was then resuspended in 990 µL of DMEM with 10% FBS. The cells were treated with these samples then incubated for 48 h. Cell viability was then measured using an MTT assay (described below). DMEM media was used as a control. Samples were measured as triplicates and a mean value was calculated.

### 2.13. Measuring Toxicity of Doxorubicin-Bound MgF_2_ Particles by MTT Assay

Cells were cultured and seeded as mentioned previously, then treated with nanoparticle- bound doxorubicin. Particles were formed using 10 µL of 1 M NaF added to 20 µL of 1 M MgCl_2_. Dox drugs of 100, 200, 300, 400 and 500 nM concentrations were used. The treated cells were incubated for 48 h. Cell viability was then measured using an MTT assay, as described below. Plain DMEM media, as well as unbound Dox of concentrations 100, 200, 300, 400, 500 nM were used as controls. All samples were measured as triplicates and a mean value was calculated. 50 µL of MTT solution was added to each well. The wells were then incubated in 5% CO_2_ and 37 °C for 4 h. After incubation, formazan crystals were formed in the wells. The solutions in the wells were discarded and 300 µL DMSO was used to dissolve the formazan crystals (over 5 min). The dissolved formazan solutions were then measured for absorbance, using a spectrophotometer (Bio-Rad Microplate Reader, Hercules, CA, USA) at wavelength of 595 nm and reference wavelength 630 nm.

The samples were prepared as triplicates and a mean was calculated for each. To measure the cell viability as a percentage, the absorbance value of each sample was divided by the absorbance value for the control (media only), and multiplied by 100.

### 2.14. Protein Corona Analysis Using Liquid Chromatography and Mass Spectrometry (LC/MS)

20 µL of 1 M of MgCl_2_ was added to 10 µL of 1 M NaF and incubated at 37 °C for 30 min. DMEM media supplemented with 10% FBS. The final volume the sample was 1 mL. The sample was then incubated again at 37 °C for 30 min then centrifuged at 13,000 rpm and 20 °C for 10 min. 970 µL of the supernatant was then removed and replaced with 970 µL water, without mixing with the precipitate. The sample was centrifuged a second time under the same settings, and 970 µL of supernatant was discarded. The precipitate was resuspended in 100 µL of 50 mM EDTA.

To perform liquid chromatography, a C18 spin column (ThermoScientific) was used. The column was centrifuged at 1500 rcf for 1 min after each solution was added in order for it to flow through. Each step mentioned below was repeated twice. 200 µL of Activation solution (50% methanol) was used to rinse the column and wet the resin. This was followed up by running 200 µL of Equilibration solution (0.5% trifluoroacetic acid in 5% acetonitrile) through the column. The nanoparticle sample was mixed with ⅓ Sample Buffer (2% trifluoroacetic acid in 20% acetonitrile) and the total volume was made out to be 120 µL. The sample was run through the column, and the collected flow through was reintroduced into the column to maximise sample loading onto the resin. Next, 200 µL of Wash solution (same composition as Equilibration solution) was run through the column. 20 µL of Elution buffer (70% acetonitrile) was run, and the collected flow through (40 µL for each sample) was dried in a vacuum evaporator (Eyela centrifugal evaporator) overnight.

#### In-Solution Digestion

The pellet formed from the evaporated sample was resuspended in 100 µL. 50 µL was drawn out for further experimentation. It was added to 25 µL ammonium bicarbonate, 25 µL TFE (trifluoroethanol), and 1 µL of DTT (dithiothreitol). The sample was then incubated at 60 °C for 60 min. After that, 4 µL of IAM (iodoacetamide) was added and the sample incubated at room temperature under light sensitive conditions for 60 min. Next, 300 µL of water, 100 µL ammonium bicarbonate and 10 µL trypsin were added and the solution was incubated at 37 °C overnight. 1 µL of formic acid was then added and the sample was put in an incubator for 24 h.

The sample was subjected to mass spectroscopy and the result was analysed by protein identification through automated de novo sequencing using PEAKS software.

### 2.15. Statistical Analyses

Error bars on all the charts represent the standard deviation, which was calculated using the standard deviation formula:(1)$$\mathrm{Error}\;\mathrm{bars}\;=\;\surd \frac{\Sigma {\left(x-\bar{x}\right)}^{2}}{n}$$

GraphPad software (San Diego, CA, USA) was used to calculate *p* values for the particle cytotoxicity and cell viability results of drug (control) and particle-bound drug samples. A *t*-test was applied to compare each nanoparticle formulation’s toxicity with the control. *t*-tests were also applied to each individual nanoparticle-bound doxorubicin concentration vs the control. GraphPad software was also used to calculate the confidence interval of cytotoxicity of particle-bound drugs.

## 3. Results

### 3.1. Fabrication and Characterization of Nanocrystals

Fabrication of nanocrystals was assessed using absorbance measurements as an indication of nanoparticle formation. As shown in Figure 2, overall increase in absorbance values was observed with increasing NaF concentrations. Absorbance values were higher at 10 mM up to 30 mM and reduced at 40 mM. This dip at 40 mM was a persistent pattern observed across all repetitions of the experiment and may be accounted for by other factors unrelated to the actual absorption. This is addressed in the discussion section. Increasing the incubation time to 60 min resulted overall in an increase in absorbance values, as Figure 3 shows. Regarding the 50 mM point in Figure 3, that is likely an anomaly. The two values for 30 min and 60 min are close enough, and it seemed that incubation stops having an effect at 40 mM point probably due to the environment being saturated with NaF.

### 3.2. Dynamic Light Scattering (DLS)

Dynamic light scattering was used to measure the size of the particles formed (Figure 4). The dip at 40 mM is consistent with the spectrophotometry pattern, and is addressed in the discussion.

The two graphs in Figure 5 below give additional details on the distribution of the particles in a suspension of 20 mM of MgCl_2_ and 10 mM of NaF. The average size here was 122 nm. The polydispersity index (PDI) of the nanoparticles was 1. This value suggests a great diversity in the sizes of the particles formed. The top-most graph displays the distribution of particles by percentage. Most of the particles (in terms of sheer numbers) fall within an average of 9 nm diameter, with a smaller percentage within 32 nm average and 190 nm average. This is consistent with the diversity observed in SEM images (Figure 8). In terms of intensity, most of the light scattering was observed for particles within 190 nm diameter, followed by 1345 nm particles and 39 nm.

### 3.3. Microscopic Observation of the Particles

Particles were viewed (Figure 6) under a brightfield 10× microscope to visualise the pattern of particle growth as the concentration of NaF used in formulating the particles was increased. There was a trend for particle size to increase with NaF concentrations 10–30 mM, then drop at 40 mM, but pick up again at 50 mM. This pattern at 40 mM is consistent with the absorbance and dynamic light scattering observations.

### 3.4. FTIR

FTIR was performed to look at the chemical bonds formed within the nanoparticles. Figure 7 displays the spectrum detected. An O-H peak was observed at 3342 cm^−1^. Peaks at 1639, 1548 and 1443 cm^−1^ suggest carbonate group from DMEM. The peak at 589 cm^−1^ corresponds to Mg-F (indexed as 601 cm^−1^). No peak was observed corresponding to Mg-Cl (indexed as 721 cm^−1^) or Na-F (indexed as 599 cm^−1^), although it may have overlapped with Mg-F absorbance.

### 3.5. FE-SEM and EDX

As shown in Figure 8, the particles formed were of a cubic shape. Their sizes ranged from 200 nm to 600 nm and 1 µm. As the images suggest, the majority of particles <200 nm, which is consistent with the dynamic light scattering size distribution graphs (Figure 5). The variety of particle sizes is consistent with the high polydispersity value of ‘1’ reported with dynamic light scattering data. Figure 9 shows EDX measured at two points on an SEM image. EDX analysis showed presence of Mg, F, Na, and Cl on the crystals.

### 3.6. pH Sensitivity of the Nanoparticles

pH sensitivity of the particles was assessed through photometric absorbance of particles formulation in media of different pH. As Figure 10 shows, reducing the pH led to a decrease in absorbance values, indicating breakdown of the nanoparticles. The graph shows a steep decline in absorbance going from 0.4584 (at pH 7.5) down to 0.0904 (at pH 7.0), implying that most of the particles dissolved as you went down from 7.5 to 7.0. The particles were all dissolved by pH 5.5.

### 3.7. Drug Binding Affinity of Particles to Doxorubicin

As Figure 11 shows, binding affinity seems to be consistent for the different concentrations of doxorubicin, with minor differences detected. At 5 µM, the binding affinity was 13.5% ± 3.45 standard deviations, which places it within the vicinity of the binding affinity at 20 µM, which was 10.7% ± 1.8 standard deviations.

### 3.8. pH Release Profile of Doxorubicin

Decrease in fluorescence intensity of doxorubicin in MgF_2_-bound formulations as you reduce media pH from 7.5 to 5.0 was used to estimate the amount of doxorubicin released from the nanoparticles. Figure 12 shows that there was a general pattern of more doxorubicin being released as pH decreased, in line with the results observed with pH sensitivity of the nanoparticles. A gradual increase was observed from 7.5 to 6.0. The highest release percentage was observed at pH of 5, where 83% of doxorubicin was released.

### 3.9. Cellular Uptake of Doxorubicin-Bound Nanoparticles

To assess the cellular uptake of MgF_2_-bound doxorubicin, MCF-7 cells were treated with 10 µM particle-bound doxorubicin and 10 µM free doxorubicin as a control. Media was removed and the wells were rinsed with PBS, before examination under a fluorescence microscope. Fluorescence was measured after two hours and was more pronounced in cells receiving particle-bound doxorubicin than the free drug alone, thereby indicating a higher uptake of MgF_2_-bound drug compared to free drug, as Figure 13 shows.

### 3.10. Cytotoxicity of MgF_2_ Particles

The toxicity of MgF_2_ particles was assessed by calculating the percentage of cell viability for the different formulations of MgF_2_. These percentages were based on the results of MTT assays (see methods). As Figure 14 shows, there was an overall decrease in cell viability with increasing NaF concentrations. A cell viability of 80% (*p* = 0.0414) was observed with 10 mM NaF. As a result, 10 mM of NaF was chosen for further experimentation. Cell viability at 20 was 24% (*p* = 0.0002), 30 was 8% (*p* < 0.0001), 40 was 12% (*p* < 0.0001), 50 was 22% (*p* < 0.0001).

### 3.11. Toxicity of Doxorubicin-Bound MgF_2_ Particles

Toxicity of free doxorubicin was compared with MgF_2_-bound doxorubicin of the same concentration (a range of 100, 200, 300, 400, 500 nM was used). Just like with nanoparticle toxicity, MTT assays were used to calculate cell viability percentages. As Figure 15 shows, cell viability decreased in both control and treatment as doxorubicin concentration increased. MgF_2_-bound doxorubicin had lower cell viabilities compared with free drug in most cases.

At 100 nM, cell viability was 61.5% (*p* < 0.0001; 95% confidence interval (CI): 54.6–68.4) for control, and 48.4% (*p* = 0.0044; 95% CI: 10.2–86.7) for treatment. At 200 nM, it was 43.0% (*p* = 0.0012; 95% CI: 13.1–72.8) for control, and 33.3% (*p* = 0.0001; 95% CI: 14.3–52.2) for treatment. At 300 nM, it was 34.8% (*p* = 0.0002; 95% CI: 14.1–55.3) for control, and 30.2% (*p* < 0.0001; 95% CI: 16.0–44.3) for treatment. At 400 nM, it was 23.6% (*p* < 0.0001; 95% CI: 19.3–28.1) for control, and 26% (*p* < 0.0001; 95% CI: 19.2–32.8) for treatment. At 500 nM, it was 14.2% (*p* < 0.0001; 95% CI: 10.5–17.8) for control, and 18.7% (*p* < 0.0001; 95% CI: 14.6–22.8) for treatment.

### 3.12. Protein Corona Analysis Using Liquid Chromatography and Mass Spectrometry (LC/MS)

Table 1 shows the constituents of the protein corona around the nanoparticle in 10% FBS. Most of the proteins bound to MgF_2_ particles have an isoelectric point <7.5, i.e., are negatively charged in physiological pH. They would most likely be drawn to the positive region around the Mg^2+^ ions in the nanoparticle structure. A few proteins, like the keratins and globins, are positive in physiological pH and are likely drawn to the F^−^ ions in the nanoparticle structures. There is a great variety in the molecular weight of the proteins in the corona.

Figure 16 is a pie chart that shows the distribution of proteins in the corona by functions. As the figure showed, transport proteins made up the largest segment (57%) of the protein corona. This included albumen proteins, which are the most abundant proteins in plasma, constituting around 55% of plasma proteins [10], and are known dysopsonins, i.e., inhibit opsonisation. Opsosnins such as globulins, complement proteins and fibrinogen were missing in the corona [11].

## 4. Discussion

Drug delivery vectors can reduce the toxicity of anti-cancer drugs by targeted delivery to tumour tissue and reduction in off-target effects. Previous studies have looked into using inorganic vectors such as calcium phosphate [12], calcium carbonate [13] and magnesium silicate [14] for drug delivery. As of yet, no conclusive developments have been made in the field of inorganic nanocrystal drug carriers. Our study aimed to further our understanding of the capabilities of inorganic nanocrystals, by experimenting on magnesium fluoride. We assessed the presence (or lack of presence) of several important features, such as size, shape, composition, binding affinity, pH-based controlled release, internalisation into cells, safety of the nanoparticles, and enhancement of the toxicity of doxorubicin.

We have demonstrated the capability of producing MgF_2_ nanocrystal particles by precipitating MgCl_2_ and NaF. The particles formed were size-modifiable; the spectrophotometry, zeta sizer analysis and microscopic observation showed increasing size with increasing concentration of NaF. Being able to modify the size of the particles is crucial to drug delivery, as size is a major determinant in clearance from the body, uptake into tumour tissue (by enhanced permeation and retention effect (EPR), where particles are small enough to pass through leaky tumour vessels but too large to leak out of regular vessels) and internalisation into cancer cells. There is no consensus on the ideal size of nanoparticles, but close estimates have been made in the range of 10 nm to 200 nm [15]. Particles that are too small (8 nm or less) could be cleared by the kidney [16], whereas larger particles, from 200 nm up to 10 µm are more likely to be cleared by the reticuloendothelial system and less likely to be taken up by cells [15,17,18].

There was an observed decrease in absorbance, z-average diameter, and microscope observation when particles were formed using 40 µL of NaF. This observation was consistent in all the experiments and the cause of it is unknown. An experiment like this has not been done before, so there lacks any data or plausible explanation for this peculiarity. As a result of these investigations, we decided to use the 20 µL of 1 M MgCl_2_ and 10 µL of 1 M NaF formulation for the remaining research, as it had the lowest value using the lowest concentrations, and was less likely to aggregate.

FTIR was used to determine the composition of the particles. Predictable peaks were observed corresponding to carbonate, O-H and Mg-F bonds. The measuring device was less capable of detecting specific absorbance peaks for the ionic bonds in the salts.

EDX analysis was another test of composition (performed live with SEM). It did show traces of Na and Cl_2_ in the crystals, displaying a variety in the composition of the crystals. This lack of ‘purity’ is to be expected in a simple precipitation method.

SEM was used to assess the shape and diameter of the particles formed using 20 µL of 1 M MgCl_2_ and 10 µL of 1 M NaF. SEM analysis showed the shape of these crystals to be cubic in nature. What effect the shape of a particle has on its function is still not a settled topic. Work by Ye et al. suggests that non-spherical nanoparticles (such as in this case) are more resistant to sequestration by the reticuloendothelial system [19]. The diameter ranged from 200 nm–1 µm. Most of the particles were, however, smaller than 200 nm (seen in the background of the images). This is consistent with the size distribution graph (Figure 5) and the high polydispersity value reported by the zetasizer software reported for this formulation.

Drug binding to MgF_2_ was consistent for the different concentrations of doxorubicin we tested: 5, 10, 15 and 20 µM. The binding affinity is comparable to that of carbonate apatite; reported by Hossain et al. as 19% for 5 µM of doxorubicin [20] (our value for 5 µM of doxorubicin was 13.5% ± 3.45, giving us an upper limit of 17%). It could be improved by modifications to MgF_2_, e.g., conjugation with ligands.

pH sensitivity of the nanoparticles is important for a controlled release function. Together with a pH release profile of doxorubicin, we could have a picture of whether MgF_2_ nanoparticles can be relied upon to release doxorubicin only when in (more acidic) tumour tissues. pH sensitivity was observed by comparing spectrophotometric absorbance values of particles formed in media of pH 7.5, 7.0, 6.5, 6.0, 5.5. pH sensitivity results suggest that most of the nanoparticles will have dissolved at 7.0 pH, which is within the range of pH in a tumour microenvironment (believed to be in the range of 6.8–7.2, although the values are not consistent throughout the tumour tissues) [21]. Further modifications may have to be made to modulate the sensitivity of MgF_2_ to pH.

The experiment on pH associated release of doxorubicin produced a gradual curve upwards on the % release vs pH graph. This is consistent with pH sensitivity, as you would expect more doxorubicin to be released as more particles breakdown as the pH decreases.

The fluorescence of cells treated with MgF_2_-bound doxorubicin was compared with the fluorescence of cells treated with free unbound drug (the control) to estimate if cellular uptake of doxorubicin was changed when bound to MgF_2_. The increased fluorescence of cells treated with MgF_2_-bound doxorubicin indicated that MgF_2_-bound doxorubicin was more readily taken up into cancer cells than the free drug. Cellular uptake depends on various factors like particle charge and hydrophilicity/hydrophobicity [22], but size is the most important one [15]. Particles in the range of 60–200 nm in size can be taken up through the clathrin pathway. Particles of 200 nm–1 uM can be taken up through the caveolae pathway [22]. Considering the size of the MgF_2_ particles, they would most likely have been taken up by the clathrin-mediated pathway.

MTT analysis showed low cytotoxicity of nanoparticles in low concentrations. When using 10 µL of NaF, the particles formed reduced cell viability by 20% when compared to media alone. Our findings are consistent with other studies on magnesium-based nanoparticles, which reported low levels of toxicity [23,24,25]. MTT analysis was also used to determine the effect of MgF_2_ binding on the toxicity of doxorubicin. We observed enhanced toxicity of doxorubicin when bound to MgF_2_ particles. Optimum reduction in cell viability was observed at 400 nM concentration of doxorubicin, where we brought down cell survival to under 30%. The enhanced toxicity of doxorubicin can be attributed to enhanced cellular uptake, as observed in the previous experiment.

A protein corona forms naturally around any object inserted into plasma/serum, and generally comprises protein constituents of the bodily fluid. The types of proteins that attach to a nanoparticle depend on its size, charge and solubility. Protein corona formation can affect the size of the particles, its clearance by the reticuloendothelial system and its cellular uptake.

The most abundant proteins in the serum/plasma are albumin proteins, followed by immunoglobulins and coagulation proteins. The constituents of the protein corona, shown in Figure 16 and Table 1, display an abundance of transport proteins, including several albumin proteins, but a lack of immunoglobulins. Some coagulation factors and acute phase reactants are present, which may contribute to opsonisation.

Opsonisation is the process of ‘tagging’ foreign bodies with opsonins to facilitate clearance by the reticuloendothelial system. Opsonins are mostly immunoglobulins, complement proteins and fibrinogen. On the other hand, there are molecules in the plasma which protect foreign molecules from clearance by the reticuloendothelial system. These include albumins and apolipoproteins, both of which were observed in our nanoparticle protein corona.

In conclusion, the nanoparticles we fabricated in 10% FBS-supplemented DMEM formed a unique protein corona consisting mainly of transport proteins with lack of presence of common opsonins. This suggests that these nanoparticles will be less likely to be cleared out by the reticuloendothelial system and thus would have an increased half-life in plasma. Limitations concerning protein corona would relate to its effect on increasing the particle size, which would undermine the EPR effect crucial for accumulation of the particles in target tissues [11,26].

In summary, the benchwork and in vitro experiments displayed a potential for MgF_2_ particles to be used as a drug delivery system. Modifications to the nanocrystals will be needed to improve drug binding affinity and cellular uptake. The efficacy of the particles in vivo is yet to be assessed, as is the capability of surface modification with ligands to improve specific targeting and avoid sequestration in the liver and reticuloendothelial systems.

## 5. Conclusions

Off-target side effects of anti-cancer drugs increase comorbidity of breast cancer patients and overall healthcare cost. These issues could be addressed by targeted delivery using drug delivery systems. In this study we looked into the possibility of using inorganic MgF_2_ nanocrystal particles as a drug delivery vector. The fabrication of nanoparticles was simple and non-toxic at 10 mM concentration. The particles were size-modifiable and capable of binding doxorubicin and internalising into cells. We demonstrated their sensitivity to pH and were thus capable to control their release of the drug by reducing the pH. The particle-bound doxorubicin formulation enhanced the effect of doxorubicin in killing cancer cells.

## Figures and Tables

**Figure 1 toxics-07-00010-f001:**
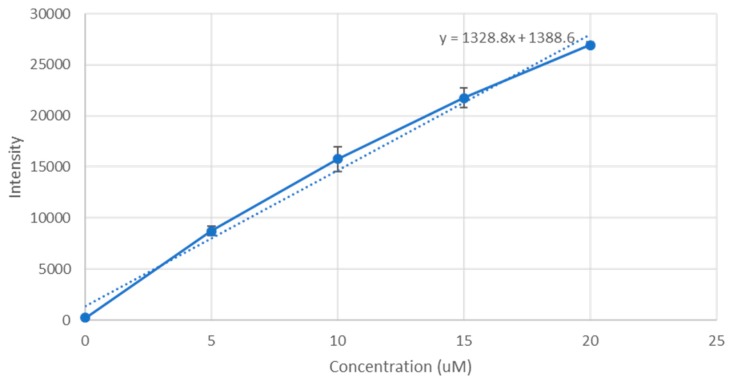
A standard graph of fluorescence intensity vs concentration of doxorubicin. Doxorubicin was prepared in concentrations of 0, 5, 10, 15 and 20 µM in PBS. The samples were placed in 96 well plates and measured for fluorescence intensity using PerkinElmer microplate reader. The excitation wavelength was set to 495 nm and emission wavelength was set to 535 nm. The results were plotted into a scatter graph. Regression analysis of the results was used to construct a straight line graph with the formula *y* = m*x* + c, where ‘*y*’ is the fluorescence intensity measure and ‘*x*’ is the concentration of doxorubicin. *R*^2^ = 0.990441.

**Figure 2 toxics-07-00010-f002:**
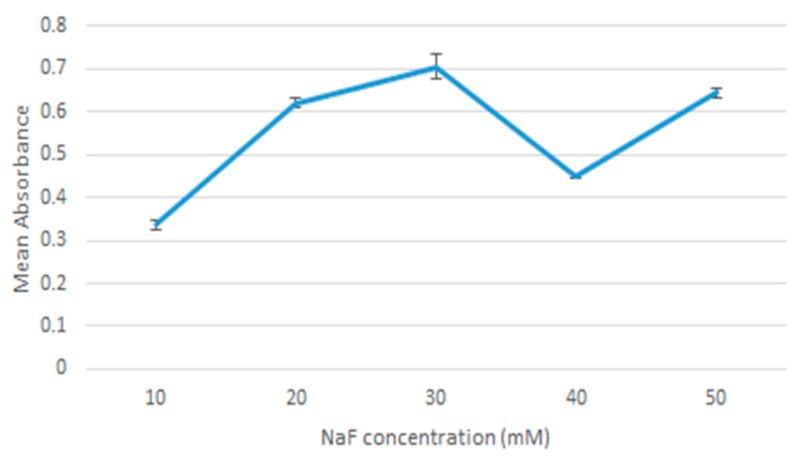
Absorbance values of MgF_2_ measured against 10 μL increments of NaF.

**Figure 3 toxics-07-00010-f003:**
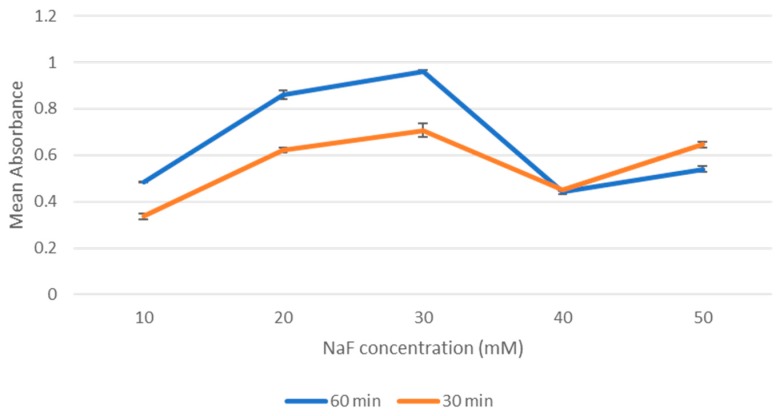
The effect of incubation time on absorbance values.

**Figure 4 toxics-07-00010-f004:**
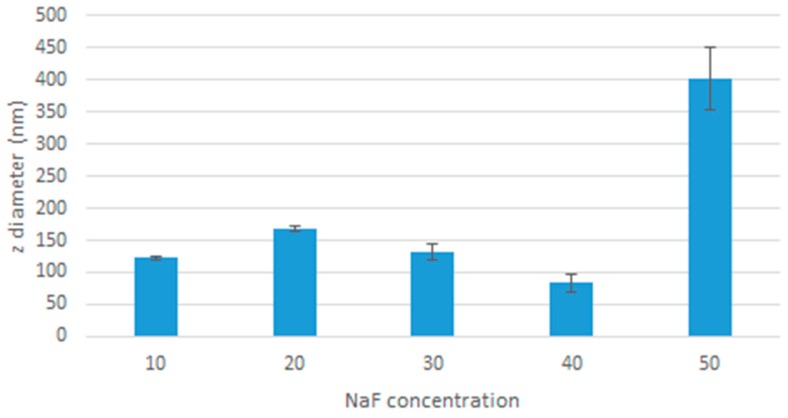
Effect of increased NaF concentration on the average diameter of MgF_2_ particles.

**Figure 5 toxics-07-00010-f005:**
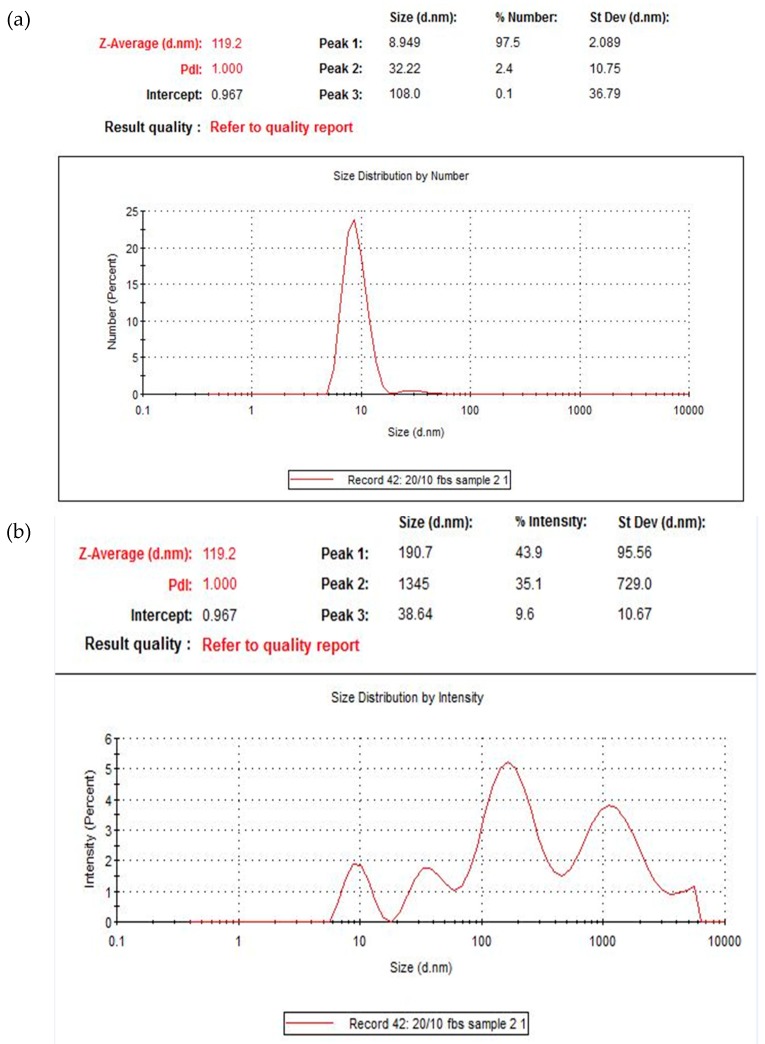
Size distribution of MgF_2_ particles in a suspension of 20 mM of MgCl_2_ and 10 mM of NaF in terms of a) % numbers b) % intensity.

**Figure 6 toxics-07-00010-f006:**
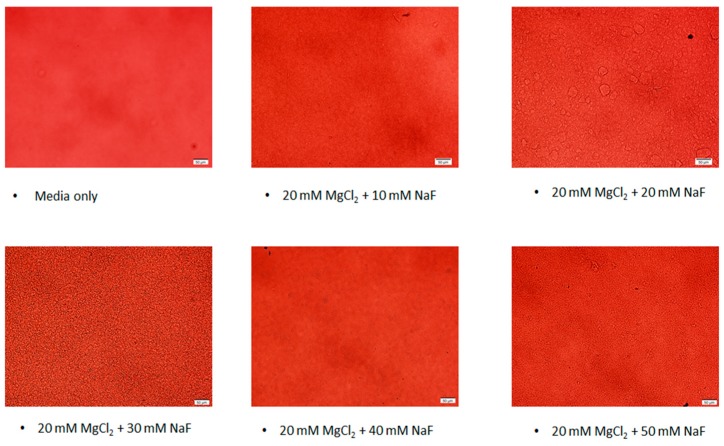
Microscopic images of the MgF_2_ suspensions.

**Figure 7 toxics-07-00010-f007:**
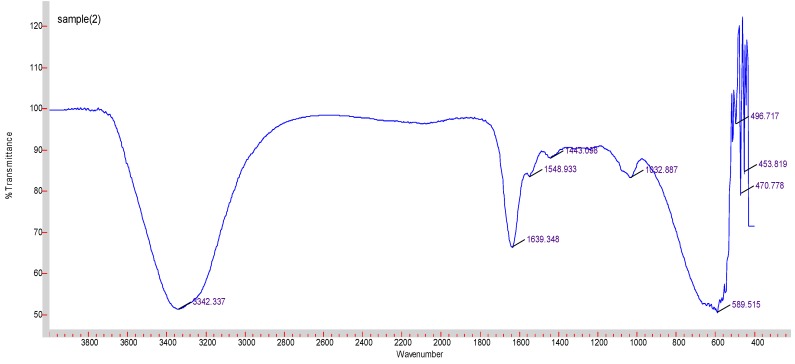
FTIR spectrum graph for MgF_2_.

**Figure 8 toxics-07-00010-f008:**
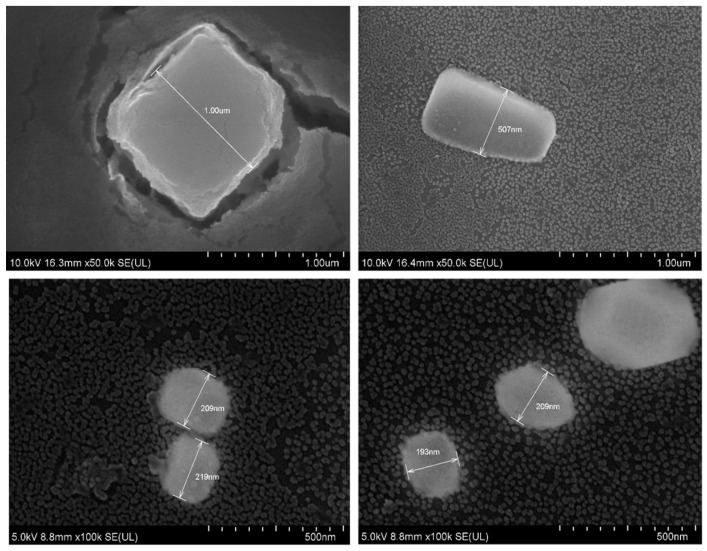
SEM images of MgF_2_.

**Figure 9 toxics-07-00010-f009:**
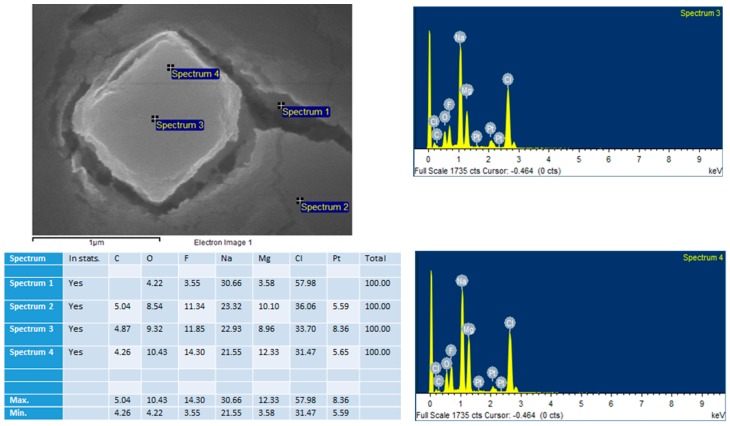
An SEM image of MgF_2_ with EDX measured at two points on the image: ‘Spectrum 3’ and ‘Spectrum 4’.

**Figure 10 toxics-07-00010-f010:**
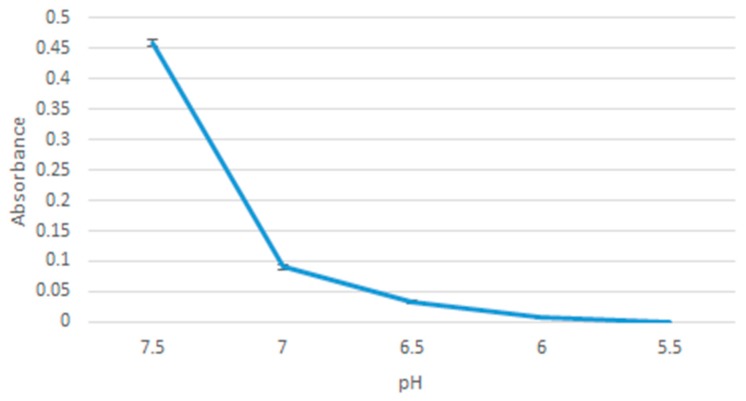
Absorbance of MgF_2_ measured against decreasing pH values.

**Figure 11 toxics-07-00010-f011:**
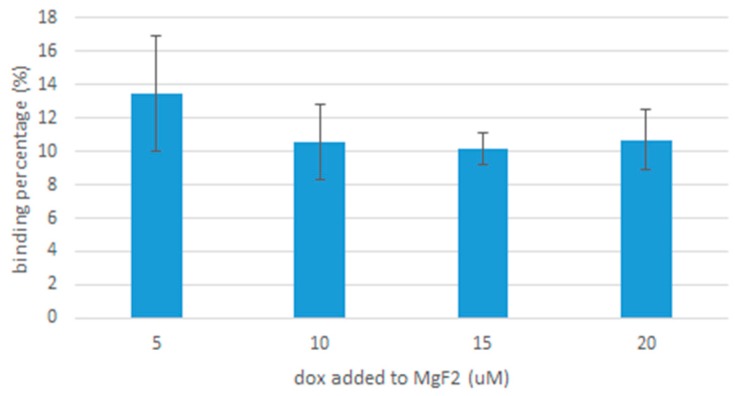
The binding affinity of different concentrations of doxorubicin bound to MgF_2_.

**Figure 12 toxics-07-00010-f012:**
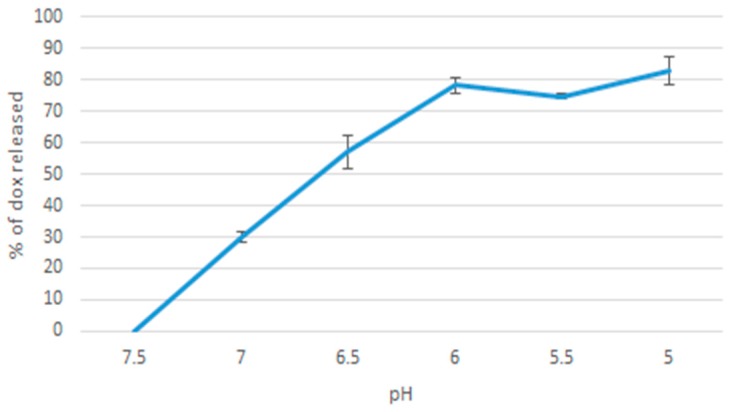
% doxorubicin released as pH was decreased.

**Figure 13 toxics-07-00010-f013:**
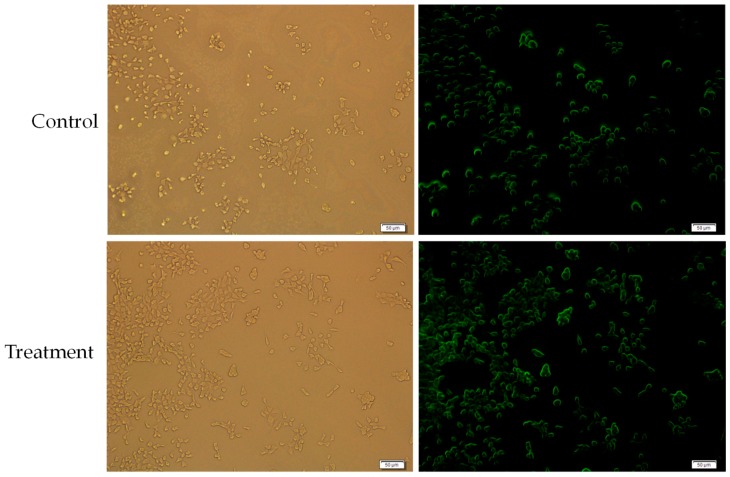
Cellular uptake measured after 2 h of control (doxorubicin as a free drug) and treatment (doxorubicin bound to MgF_2_ nanoparticles).

**Figure 14 toxics-07-00010-f014:**
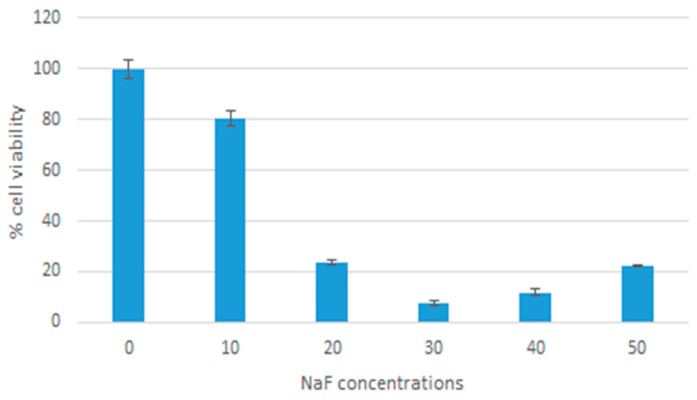
Cytotoxicity of MgF_2_ nanoparticles.

**Figure 15 toxics-07-00010-f015:**
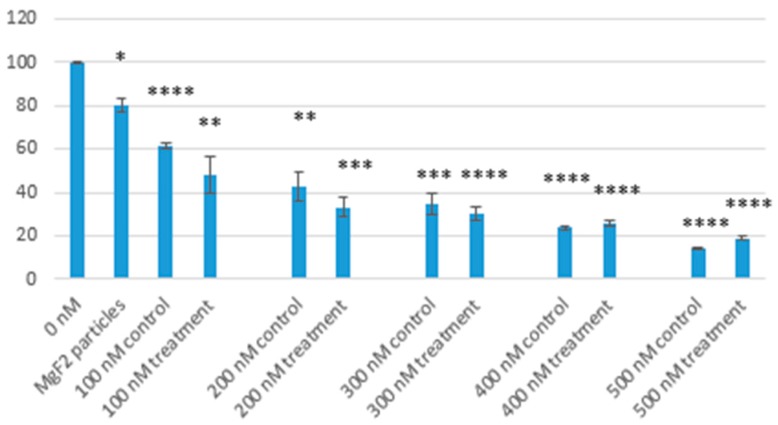
Comparing cell viabilities of MgF_2_ nanoparticle, doxorubicin-bound MgF_2_ particles and free drug controls. Control = drug in media; Treatment = particle-bound doxorubicin, as described; * = *p* value 0.05–0.01; ** = *p* value 0.01–0.001; *** = *p* value 0.001–0.0001; **** = *p* value < 0.0001; *p* values were calculated using 0 nM for comparison.

**Figure 16 toxics-07-00010-f016:**
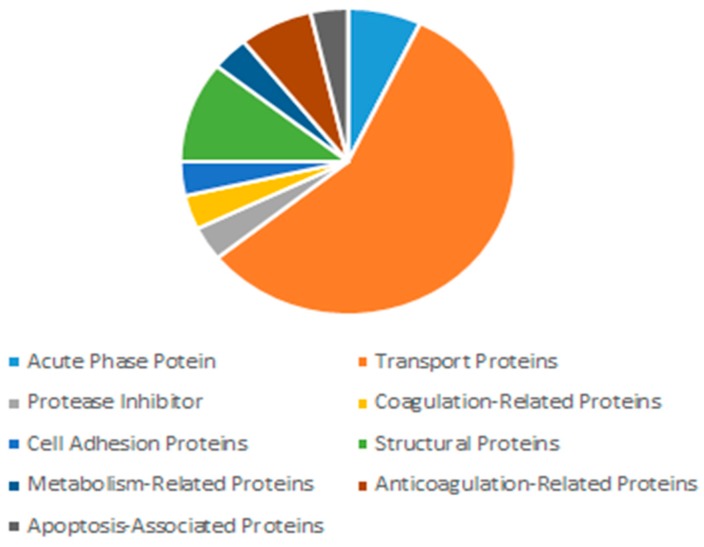
Distribution of proteins in the corona by function in 10% FBS.

**Table 1 toxics-07-00010-t001:** Protein corona constituents formed around the nanoparticles in 10% FBS-supplemented DMEM media.

No.	Description	Avg. Mass	Coverage (%)	Function	Isoelectric Point (pI)
1	Alpha-2-HS-glycoprotein	38,419	85	acute phase protein	5.94
2	Alpha-2-HS-glycoprotein	38,419	85	acute phase protein	5.94
3	ALB protein	69,294	73	transport and binding protein	5.4
4	Serum albumin	69,324	72	transport and binding protein	5.4
5	Serum albumin	69,294	72	transport and binding protein	5.4
6	Apolipoprotein A-I	30,276	69	lipid transport	5.56
7	Apolipoprotein A-I preproprotein	30,276	69	lipid transport	5.56
8	Alpha-1-antiproteinase	46,104	24	protease inhibitor	5.4
9	Prothrombin	70,506	18	coagulation	5.64
10	Vitronectin	53,575	19	cell adhesion	5.55
11	Apolipoprotein A-II	11,202	60	lipid transport	5.56
12	Hemoglobin fetal subunit beta	15,859	39	oxygen transport	7.1
13	Keratin 1	63,151	9	structural protein	8.15
14	Keratin type I cytoskeletal 10	54,848	19	structural protein	8.15
15	Keratin 10	54,849	19	structural protein	8.15
16	Adiponectin	26,133	20	fat metabolism	5.42
17	Serotransferrin	77,753	14	iron transport	6.81
18	Serotransferrin	77,666	14	iron transport	6.81
19	Globin C1	15,184	44	oxygen transport	6-8
20	Hemoglobin subunit alpha	15,184	44	oxygen transport	7.1
21	Antithrombin-III	52,440	9	anticoagulant	5.93
22	Antithrombin-III	52,347	9	anticoagulant	5.93
23	Vitamin D-binding protein	53,356	7	vitamin d transport	5.2
24	Vitamin D-binding protein	53,328	7	vitamin d transport	5.2
25	Vitamin D-binding protein	53,342	7	vitamin d transport	5.2
26	APOM protein	13,027	17	lipid transport	5.66
27	Apolipoprotein M	21,158	11	lipid transport	5.66
28	Apoptotic chromatin condensation inducer 1	150,755	1	apoptosis	6.08

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
