# Peer review of "Magnesium Fluoride Forms Unique Protein Corona for Efficient Delivery of Doxorubicin into Breast Cancer Cells"

_toxics, 2019, doi:10.3390/toxics7010010_

Round 1
Reviewer 1 Report
The paper presents results of studies on MgF2 nanoparticles for doxorubicin delivery. Some points should be explained before publication:
1. Fig 1 - R2 should be added to show the correlation
2. The size distribution of nanoparticles is very high. Fraction of the largest nanoparticles cannot be administered by injection. Is a sieving or another method considered for separation of large particles? Were the particles filtered for in vitro experiments on MCF-7 cells.
3. Figure 8, 9 – the scale bars are invisible
4. Conclusion – Authors claim that „The fabrication of nanoparticles was simple and non-toxic”. However, according to the results the final nanoparticles were non-toxic only with 10mM of NaF. It should be clarified.
Author Response
Reviewer 1:
The paper presents results of studies on MgF2 nanoparticles for doxorubicin delivery. Some points should be explained before publication:
1. Fig 1 - R2 should be added to show the correlation
2. The size distribution of nanoparticles is very high. Fraction of the largest nanoparticles cannot be administered by injection. Is a sieving or another method considered for separation of large particles? Were the particles filtered for in vitro experiments on MCF-7 cells.
3. Figure 8, 9 – the scale bars are invisible
4. Conclusion – Authors claim that „The fabrication of nanoparticles was simple and non-toxic”. However, according to the results the final nanoparticles were non-toxic only with 10mM of NaF. It should be clarified.
Response:
1. We have added the R2 to Figure 1 as suggested
2. This is correct; a fraction of the largest nanoparticles will be too large to accumulate in cancer tissues by EPR. This was a limitation to the efficacy of our nanoparticle formulation. In this particular study we didn’t use sieving or filtering methods to remove these large particles due to the tendency of the particles to further aggregate in absence of serum with increasing time.
3. We increased the size of the images in Figures 8 and 9 and the scale bars are now visible
4. We have added the clarification as suggested.
Reviewer 2 Report
This manuscript developed MgF2 nanocrystals used for the delivery of anti-cancer drugs such as doxorubicin. The authors profiled and characterized the nanocrystals by absorbance, dynamic light scattering, microscopic observance, pH sensitivity test, SEM, EDX, and FTIR. The binding affinity of nanocrystals to doxorubicin indicating they can be used as good vehicles to load anti-cancer drugs. In addition, cellular assay including uptake and toxicity was also evaluated for these nanocrystals. It is interesting that MgF2 nanocrystals were pH sensitive, which would benefit for delivery of drugs to acidic tumor tissues. Some detailed comments are as follows.
1. Quality of figures in this manuscript need to be improved. Resolutions of Figure 6, 7, 9 and 13 should be improved. Error bar should be added in Figure 1. Regarding to Figure 2 and 3, the title of y axis should be consistent.
2. Accuracy. Carefully read and proof is needed.
Line 199. NaCl should be NaF.
Line 248-250, “at 30 and 40 mM, there were peaks in the size distribution graph at higher diameters than observed at 10 and 20 mM”. But from the data shown in Figure 4, I cannot see the higher diametes in 30 mM or 40 mM group. What’s the calculation for the error bar, SD or SEM? This should be notified in the method part.
3. Figure 2 and 3. It is interesting that 40mM causes a decrease for absorbance. More discussion should be addressed in the discussion part. For 50mM point, the 30 min group is higher, that is different from other concentration point. What is the possible reason? Additionally, in figure 14, lower concentration of NaF, lower toxicity is observed. And the authors finally choose 10mM in the following study. Have you tried concentrations lower than 10mM? One or two concentration points below 10mM should be investigated for the crystal characterization and toxicity assay. Similar, high concentration above 50mM is better to be profiled to confirm the absorbance result.
4. Figure 11, what is the clinical concentration of doxorubicin? This should be included in the study.
5. Figure 13. The doxorubicin concentration was demonstrated by fluorescence. Another method like measurement of intracellular doxorubicin by LC/MS is better to be provided, which will be more accurate to evaluate the uptake efficiency.
line 317, “10 μM particle-bound doxorubicin” the 10 uM here is before or after loading by MgF2 crystal?
6. Figure 15. What are the groups calculated for the p values? Compared with 0 nM group or MgF2 particles group? This should be described.
7. pH sensitive is an interesting point for this system. It is better to test in mice showing that MgF2 crystal is good for delivery of doxorubicin to tumors. Or at least another tumor cell line should be tested for this property.
Author Response
Reviewer 2:
This manuscript developed MgF2 nanocrystals used for the delivery of anti-cancer drugs such as doxorubicin. The authors profiled and characterized the nanocrystals by absorbance, dynamic light scattering, microscopic observance, pH sensitivity test, SEM, EDX, and FTIR. The binding affinity of nanocrystals to doxorubicin indicating they can be used as good vehicles to load anti-cancer drugs. In addition, cellular assay including uptake and toxicity was also evaluated for these nanocrystals. It is interesting that MgF2 nanocrystals were pH sensitive, which would benefit for delivery of drugs to acidic tumor tissues. Some detailed comments are as follows.
1. Quality of figures in this manuscript need to be improved. Resolutions of Figure 6, 7, 9 and 13 should be improved. Error bar should be added in Figure 1. Regarding to Figure 2 and 3, the title of y axis should be consistent.
2. Accuracy. Carefully read and proof is needed.
Line 199. NaCl should be NaF.
Line 248-250, “at 30 and 40 mM, there were peaks in the size distribution graph at higher diameters than observed at 10 and 20 mM”. But from the data shown in Figure 4, I cannot see the higher diameters in 30 mM or 40 mM group. What’s the calculation for the error bar, SD or SEM? This should be notified in the method part.
3. Figure 2 and 3. It is interesting that 40mM causes a decrease for absorbance. More discussion should be addressed in the discussion part. For 50mM point, the 30 min group is higher, that is different from other concentration point. What is the possible reason? Additionally, in figure 14, lower concentration of NaF, lower toxicity is observed. And the authors finally choose 10mM in the following study. Have you tried concentrations lower than 10mM? One or two concentration points below 10mM should be investigated for the crystal characterization and toxicity assay. Similar, high concentration above 50mM is better to be profiled to confirm the absorbance result.
4. Figure 11, what is the clinical concentration of doxorubicin? This should be included in the study.
5. Figure 13. The doxorubicin concentration was demonstrated by fluorescence. Another method like measurement of intracellular doxorubicin by LC/MS is better to be provided, which will be more accurate to evaluate the uptake efficiency.
line 317, “10 μM particle-bound doxorubicin” the 10 uM here is before or after loading by MgF2 crystal?
6. Figure 15. What are the groups calculated for the p values? Compared with 0 nM group or MgF2 particles group? This should be described.
7. pH sensitive is an interesting point for this system. It is better to test in mice showing that MgF2 crystal is good for delivery of doxorubicin to tumors. Or at least another tumor cell line should be tested for this property.
Response:
1. Error bars were added to Figure 1, the title of the y axis was fixed for Figures 2 and 3. We increased the size of the images in Figure 13 in order to improve the resolution. We can provide the original images for Figures 6, 7, 9 and 13 as supplemental material for readers who want better resolution.
2. Line 199 (now 203) has been fixed; Lines 248-250 we have omitted because it is too nuanced and confusing. It basically relates to size distribution data for 30 mM and 40 mM, like the ones in Figure 5. There is no space to upload two graphs for each sample (that would make 10 graphs in total and could be confusing to the reader to compare side by side); regarding error bars and standard deviation, I have clarified that in the methods (section 2.15) as requested.
3. The decrease associated with 40 mM has been addressed in the discussion. It is a real observation that has been spotted over a dozen times across different experiments. We have been unable to find any rationale behind it. Since a research into MgF2 nanoparticles hasn’t been done before, I had no previous work to cite as a reference. We can’t belabour this point as we have no substantial explanation for this.
Regarding the 50 mM point in figure 3, that is likely an anomaly. The two values for 30 minutes and 60 minutes are close enough, and it seemed that incubation stops having an effect at 40 mM point probably due to the environment being saturated with NaF. We added this to section 3.1.
Concentrations below 10 mM of NaF would produce few particles while concentration above 50 mM would generate too many particles with the consequence of much bigger size particles.
4. We think there may have been a misunderstanding here. In Figure 11, the concentration of doxorubicin is in the y axis; 5, 10, 15, 20 uM, and the % of them that bound to the nanoparticles (i.e. binding affinity) on the x axis.
5. We haven’t considered using LCMS for uptake, although the fluorescence data enabled us to assess the uptake. The technique used in LCMS was to analyse the proteins only.
Line 317 (now 345-346) and wherever ‘doxorubicin-bound nanoparticles’ are mentioned, we are referring to the process outlined in the methods section 2.11; the drug was loaded with the other reagents (20 mM MgCl2 and 10 mM NaF) during the fabrication process.
6. We added a clarification in Figure 15 that the p values used 0 nM as the comparison for calculating p values.
7. We are considering testing in mice for future studies.
Round 2
Reviewer 2 Report
It is OK after revision.